# Handwritten stroke augmentation on images

## Abstract

In this paper, we introduce Handwritten stroke augmentation, a new data augmentation for handwritten character images. This method focuses on augmenting handwritten image data by altering the shape of input character strokes in training. The proposed handwritten augmentation is similar to position augmentation, color augmentation for images but a deeper focus on handwritten character strokes. Handwritten stroke augmentation is data-driven, easy to implement, and can be integrated with CNN-based optical character recognition models. Handwritten stroke augmentation can be implemented along with commonly used data augmentation techniques such as cropping, rotating, and yields better performance of models for handwritten image datasets developed using optical character recognition methods. Our source code will be available on GitHub.

## 1 Introduction

Digitization impacts multiple fields by transforming infrastructure and providing solutions through technology. With access to technologies, digital world can provide incremental economic growth (Sabbagh et al., 2012). Technologies used for developing applications for online payments, ordering and also making historical data accessible. Due to this pandemic, there is an increase in mobile usage, online bank transactions, online purchases (Bai et al., 2021; Katz et al., 2020; Akram et al., 2021). Businesses are providing services through internet finding new ways to expand like inclusion of 3D view of houses in application for real estate, grocery delivery (Sulaiman et al., 2020; Hobbs, 2020; Hillen, 2021). Existing businesses has to be transformed digitally in order to prevent drastic reduction in sales and debt repayments to bank (Indriastuti & Fuad, 2020). If such businesses comes online, then the historic data has to be made online. Historic data refers to hard copy comprising of applications, letters and any official information used offline by the company. For example, business serving regionally may have the documents in their own language. The existing work has to be digitized and the conversion is executed using optical character recognition techniques. Optical character recognition (OCR) is essential to read paper documents and make computers understand them. When such businesses are taken online, they can serve for the regions wherever the internet is available. At the same time, businesses have to scan and read their regional languages.

Optical character recognition system has been in use since 1940s. OCR have been used for typed, handwritten characters. These systems were used for historic data conversion. Then, in 1979s, the performance and response time were improvised in systems and software were developed for commercial purposes and used till 2000s. The use of systems were to convert historic data into binary format and read metal printed text. Problems faced were non standard fonts, printing the noise (LeBourgeois, 1997). Innovations in current decade paves the way for machine learning OCR models. In recent times, OCR systems are used beyond handwritten character recognition such as document reader, signature verification in banks, forgery detection, vehicle identification. OCR system uses software which does recognition tasks (de Mello & Lins, 1999). Current OCR methods include algorithmic technique Qadri & Asif (2009), machine learning techniques such as nearest neighbour, support vector machine (Franke et al., 2002), neural networks (Pansare & Bhatia, 2012). These techniques can recognize multiple characters (Elagouni et al., 2012), multiple fonts (La Manna et al., 1999; Bharath & Rani, 2017; Samadiani & Hassanpour, 2015) , multiple size (Slimane et al., 2017) and even signature verification (Pansare & Bhatia, 2012). Current OCR systems methods can even read multiple languages and understand them. Across languages, there are handwritten documents and printed documents.

The handwritten characters are different for each language. Thus to provide a common augmentation technique is difficult without knowing the characteristics of handwriting. To develop a better augmentation technique for handwritten data, we need to know about the signature verification systems.

Signature verification is process used in banks to detect forgery. Signature verification is based on strokes in the character. The characters in signature are improperly aligned and don't have a proper structure. Such characters are verified using strokes. Every character is formed by more than one stroke. These strokes differ for each person according to apparent pen pressure, curvature of the stroke, stroke thickness, stroke width and stroke intensity (Impedovo & Pirlo, 2008; Franke et al., 2002; Pervouchine & Leedham, 2007; Pansare & Bhatia, 2012). These methods are used in forensic, banks for eliminating forgeries. Verification of signature is executed by using various methods like template matching (Pansare & Bhatia, 2012), algorithms and traditional deep learning models (Alajrami et al., 2020). Signature does have language but the verification systems serve across languages and detecting forgery. Understanding how the signature is detected and verified can help in learning about the handwriting. Signature is made by characters and characters are made by strokes. Developing augmentation technique for handwritten characters will be easier if the strokes are augmented.

There are certain data augmentation techniques are developed for specific purposes (Mushtaq et al., 2021). Such data augmentations can be used to improve the performance of the model. We propose a similar approach to develop data augmentation technique for handwritten characters. Data augmentation plays an important role in generalizing the model and improve the performance of the model. Datasets comes in various shapes and sizes. There are structured and unstructured datasets. Unstructured dataset contains incomplete data, noisy data. For larger datasets, even though the dataset is not properly structured, the model can perform better. For example, Google trillion word corpus improved the text-to-speech and text-based models. The corpus consists of unstructured, unfiltered data with errors (Halevy et al., 2009). The other smaller datasets uses data augmentation techniques to improvise the performance of model and also to generalize the model. Recent advancements also uses GANs for mimicking the dataset. In machine learning, there are very less properly documented datasets in regarding to handwritten images. Currently MNIST comprising of digits LeCun et al. (1998), KMNIST comprising of kuzushiji Japanese characters Clanuwat et al. (2018) and extended MNIST containing English alphabets are well documented datasets which are in use multiple research purposes Cohen et al. (2017). There are multilingual OCR systems similarly there has to be generalized augmentation technique for handwritten characters (Mathew et al., 2016; Aradhya et al., 2008).

The proposed augmentation technique can be used across languages for any handwritten image datasets. In this paper, we will train our own OCRNet to perform rudimentary classification for all the datasets. Then the proposed augmentation techniques are implemented to dataset while training and results are obtained. Tabulated results contain accuracy for the properly documented datasets with the imposed augmentation techniques.

## 2 Literature Review

Data augmentation for images helps in training a better accurate model. Data augmentation reduces overfitting by developing more generalized model. Especially for smaller dataset where overfitting may occur. Overfitting can also be reduced by methods like regularization, dropout. Dropout Srivastava et al. (2014) drops the node from the neural network with their connections. Similar to dropout, there is dropblock and dropconnect. Dropblock Ghiasi et al. (2018) removes a complete block of nodes in neural network. Dropconnect Wan et al. (2013) changes a randomly selected subset of weights within the network to zero. There are batch normalization which stabilize learning of neural network. There are transfer learning which can learn from the pretrained model after tuning certain parameters for a problem. These techniques increases the accuracy of the model reduces overfitting and making the neural network learn better.

Traditional data augmentation techniques for images consists of rotation, shear, flipping. There are color augmentation techniques (Shorten & Khoshgoftaar, 2019; Khalifa et al., 2021) which are used for improvising model accuracy in image datasets. Medical datasets uses specialized techniques for improving accuracy. As medical datasets are crucial and most of the datasets are private, it is important to develop a better model.

Generative adversial networks (GAN) are used in data augmentation techniques to develop closely related dataset. GAN is used in medical imaging (Han et al., 2020; Madani et al., 2018) for better improvised models than traditional data augmentation. Xiao et al. (2019) proposed a data driven augmentation technique in segmentation of images of human skin which can offer better results than the normal technique. Thus data based augmentation techniques can provide better performance of the model. The proposed approach focuses only on handwritten dataset.

Earlier signature verification are used for forensic purposes for forgery detection. Currently signature is used for personal verification in banks, order delivery conformation and verifying identity in institutions (Impedovo & Pirlo, 2008). Handwritten signatures are verified by various methods including template matching, dynamic time warping, support vector machine, (Pansare & Bhatia, 2012; Franke et al., 2002). Alajrami et al. proposes neural network for verification of signature(Alajrami et al., 2020). Using selective attention, Xiao & Leedham (1999) propose a neural network based signature verification.

## 3 Handwritten analysis

Handwritten characters consists of one or more strokes. Strokes are basic defined units on which the characters are developed. Strokes have certain length, direction, curve which are relative to the other strokes in a character. Strokes are widely used for signature verification based on pressure,velocity of each stroke. Each language have set of strokes and every person write those strokes in different way in their handwriting. This is the core function in verification of signature. Figure 1 represents the strokes used in English and Japanese letters. Strokes used for each Japanese letters ranges from 1 to 30 strokes (Tamaoka & Altmann, 2005). Thus it is important to consider the formation of letters while developing a deep learning model. From the handwritten digit dataset MNIST and Japanese kuzushiji MNIST dataset, we clearly observe the representation of stroke are thicker for certain letters. Some strokes are thin due to the varying style of handwriting to user. This acts as hindrance for model to provide much accurate analysis. The thickness and curvature aspect of strokes in letters are taken into consideration for the proposed approach.

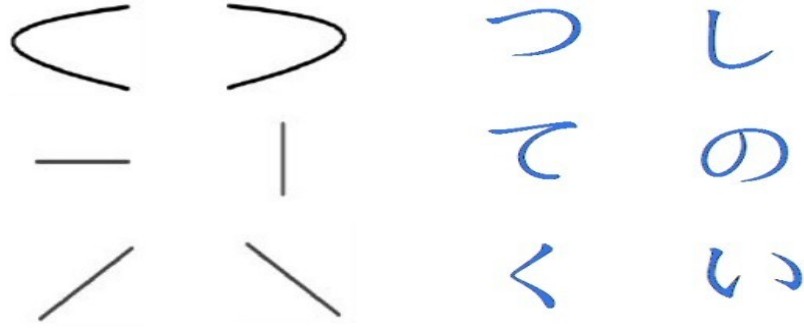

Figure 1: Image represents strokes involved in developing a character. English letter strokes are represented in black and Japanese MNIST strokes are represented in blue.

The motivation behind handwritten augmentation lies in increasing the accuracy of the model. Visually representation of the MNIST dataset, there exists certain characters where strokes should be continuous are separated. And certain characters contains less strokes when compare to original character.

## 4 Handwritten data augmentation

Strokes in the characters provide the distinctive view for a letter. Thus we try to focus on developing a variety of stroke changes in the letter. The proposed augmentation alter the strokes. The strokes in turn

alter the handwritten dataset. From the existing approaches, changes noted in signature verification software are thickness, curvature, and pressure. We incorporate the thickness aspect of the stroke in augmentation. The proposed augmentation method on handwritten character consists of thickening, thin, elongating and erasing the part of character strokes. This section elaborates the process and uses of the proposed data augmentation.

## 4.1 ThickOCR

ThickOCR is a type of augmentation where the existing strokes become bolder after this method. There are two mode in this method 'complete' and 'random'. ThickOCR method scans the pixel matrix of the PIL Image from top to bottom. Scanning is done from left to right and also from right to left for each row in the pixel matrix. In the process, the empty pixel existing before the character is altered according to the upcoming pixel with a random reduction of value in limit (0,k). The experiments are taken with k value as 10. This method is for ThickOCR 'complete' mode. The reduction of pixel value is done in order to develop a pixel which merges with the character. For 'random' mode each row in pixel matrix is scanned with a probability of 0.2 . 'Random' selects random rows and thickens the character pixel in those rows. Whereas 'complete' makes pixels in all the rows thicker. Figure 2 represents the proposed ThickOCR method for all the dataset.

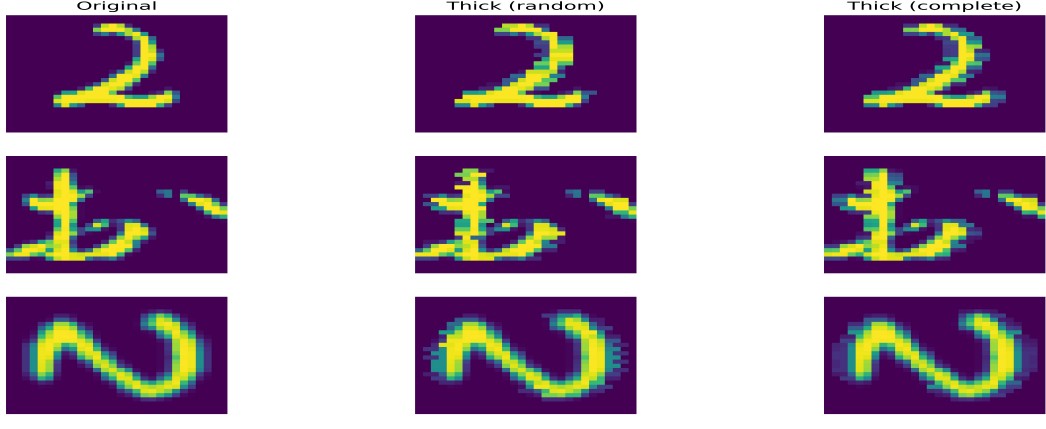

Figure 2: Thickocr

## 4.2 ThinOCR

ThinOCR method is similar to ThickOC but it erases the border character pixels to look thin. ThinOCR scan similar to ThickOCR as each row in pixel matrix is scanned from left to right and right to left. It replace the first character pixel to the current empty pixel on both the sides. ThinOCR has two mode random and complete. Complete executes for all the rows on pixel matrix. Random mode executes ThinOCR with probability of 0.2.

## 4.3 OCRElongate

OCRElongate method copies the part of the image. Expansion happens only for a single row or column resulting in a character that stays within the pixel matrix. OCRElongate method has two methods which can work on two axis horizontally x-axis or vertically y-axis. In x-axis the entire row is duplicated and for y-axis the entire column is duplicated. Figure 4 represents the elongate on x-axis and y-axis.

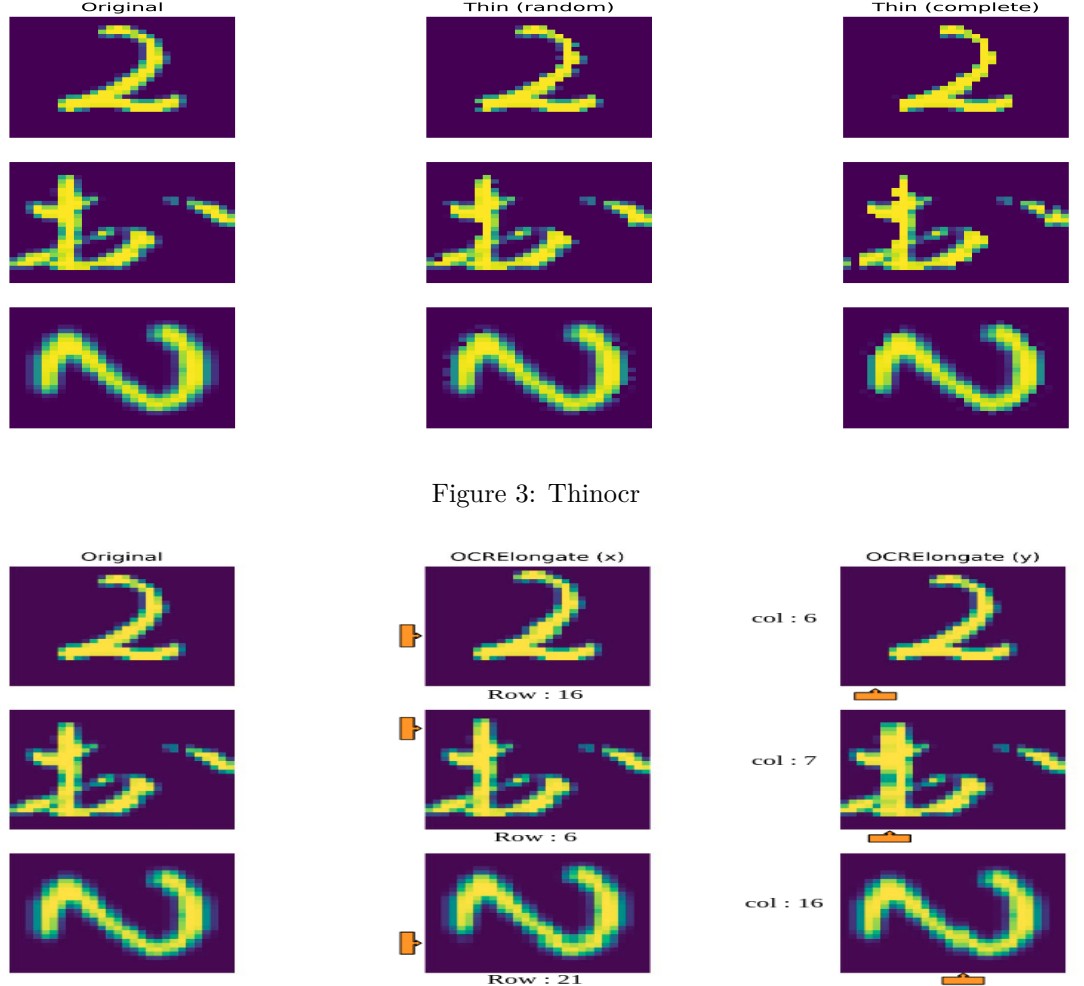

Figure 3: Thinocr

Figure 4: elongate

### 4.4 LineEraseOCR

LineEraseOCR method erases a row or column of the resultant pixel matrix. LineEraseOCR has two modes x and y. If mode is 'x', then rows are chosen and when the mode is 'y' columns are chosen randomly. The entire row/column is erased from the original image. LineEraseOCR does not alter the meaning of the image. From the figure 5, we can see that the KMNIST dataset augmented using LineEraseOCR does not entirely remove the stroke. This augmentation method removes only one row/column of the image preserving the definition of the character.

## 5 Experimentation

In this section, we evaluate the OCRNet for its capability to improve the performance of trained model on multiple tasks. We first study the datasets used, followed by the setup used for experimenting the proposed methods and then the experimented results on three chosen datasets.

**Dataset -** The dataset used are MNIST, EMNIST and KMNIST. A short detail about the dataset is added below.

MNIST - Modified National Institute of Standards and Technology dataset consists of handwritten digits used widely in research fields. This dataset contains 60000 training and 10000 test grayscale images of 28x28

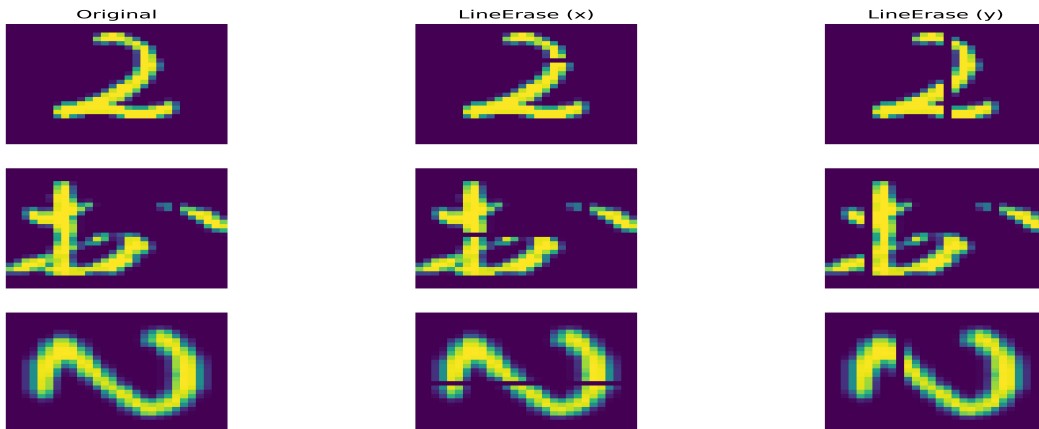

Figure 5: lineeraseocr

pixel size. This MNIST is constructed from a NIST database consisting of handwritten digits. The dataset contains 10 classes of digits from 0 to 9.

KMNIST - This is the Kuzushiji MNIST database which contains handwritten Japanese characters adapted from the Kuzushiji database. This dataset is claimed to be the drop-in replacement for MNIST dataset. Similar to MNIST dataset, this dataset contains 28x28 pixel gray scale images of 70000 in total. The dataset contains 10 classes of Japanese characters and the training set consists of 60000 images and testing set consists of 10000 test images.

EMNIST - This is an English MNIST dataset containing English alphabets. The dataset offers various categories of data. We use EMNIST letters dataset, containing 37 classes, is a mix of digits and alphabets in the class. Within alphabets there is a mix of small letter and capital letter alphabets. The dataset used with bymerge condition of letters and digits consisting of 387,361 training images and 23,941 testing images.

**Experimental setup** For testing the three augmentation methods, we run experiments on well documented datasets MNIST, Kuzhiji MNIST, EMNIST. All experiments run for 30 epochs at a learning rate of 0.0005 using 50 as batch size. The highest accuracy test accuracy at all epochs is reported as the best score.

The augmentation is performed on the input dataset with 50% probability. The augmented dataset is fed into the neural network for classification. We name this neural network as OCRNet with 2 convolutional layers paired with a max pool layer followed by a fully connected layer. This neural network is used for comparison purposes similar to SmallNet proposed by Perez & Wang (2017). One can replace the network with any network of their choice for performing classification. The detailed network is described below.

**OCRNet**
1. Conv with 10 channels and stride of 1 with ReLU activation.
2. Max pooling with window size of 2.
3. Conv with 20 channels and stride of 1 with ReLU activation.
4. Max pooling with a window size of 2.
4. Fully connected layer with an output according to dataset.

For calculating the loss, we use cross entrophy loss function. Models are trained with Adam optimizer using the learning rate as 0.001. Pytorch without GPU is used to train the model as the CPU itself is fast which needs around 2 minute per epoch.

The ThickOCR have the probability of 0.5% on input dataset and the ThinOCR also have a probability of 0.5% on input dataset. In general the proposed data augmentation methods affect 50% of the image datasets during training.

## 5.1 Experiments on MNIST

The experiment results are tabulated for MNIST dataset in Table 1. The proposed augmentation method lags behind the baseline model in terms of performance. Among them the LineEraseOCR method with x-axis have the test accuracy very close to the baseline model. The baseline for MNIST dataset provided an accuracy of 99.15% and the dropout have not been effective in the model since it is a small 3 layer model and dropping the nodes in the layer affects the model output. Among the proposed data augmentation methods ElongateOCR(y axis) is the least performed model. And the ElongateOCR among x axis performed better than the y-axis elongation. The ThickOCR(random) and ThinOCR(complete) provides the same accuracy of 98.99%. Whereas the ThickOCR(complete) performance is just above the least accurate model. The ThinOCR(random) and LineEraseOCR(x-axis) are the two model which performance is close to the baseline OCRNet model. The proposed data augmentation methods for MNIST is getting closer to the baseline model.

| Augmentation | Test Accuracy (%) |
|---|---|
| None | 99.15 |
| +Dropout(Srivastava et al., 2014) | 97.25 |
| +ThickOCR(random) | 98.99 |
| +ThickOCR(complete) | 98.93 |
| +ThinOCR(random) | 99.00 |
| +ThinOCR(complete) | 98.99 |
| +LineEraseOCR(x-axis) | 99.08 |
| +LineEraseOCR(y-axis) | 98.89 |
| +ElongateOCR(x/horizontal) | 98.92 |
| +ElongateOCR(y/vertical) | 98.87 |

Table 1: Test accuracy(%) on the handwritten image augmentation methods using MNIST dataset.

## 5.2 Experiments on Kuzushiji MNIST

The experiments on Japanese character dataset called as KMNIST is tabulated. Table 2 shows the augmentation method on KMNIST dataset. The proposed augmentation techniques have performed better than the baseline model. ElongateOCR method have better test accuracy than the other methods.

| Augmentation | Test Accuracy(%) |
|---|---|
| None | 91.48 |
| +Dropout(Srivastava et al., 2014) | 69.04 |
| +ThickOCR(random) | 91.81 |
| +ThickOCR(complete) | 91.82 |
| +ThinOCR(random) | 92.04 |
| +ThinOCR(complete) | 91.61 |
| +LineEraseOCR(x-axis) | 91.72 |
| +LineEraseOCR(y-axis) | 91.66 |
| +ElongateOCR(x/horizontal) | 92.19 |
| +ElongateOCR(y/vertical) | 92.11 |

Table 2: Test accuracy (%) on the handwritten image augmentation methods using kuzushiji MNIST dataset.

## 5.3 Experiments on EMNIST dataset

Unlike the MNIST and KMNIST datasets, EMNIST dataset consists of 37 classes. The results of the experiment is tabulated in Table 3. Augmentation 'None' represents the baseline OCRNet model.

The proposed data augmentation methods are developing better accurate models. The data augmentation methods proposed does not change the meaning of the dataset. With minor change in the handwritten

| Augmentation | Test Accuracy (%) |
|---|---|
| None | 77.99 |
| +Dropout(Srivastava et al., 2014) | 60.59 |
| +ThickOCR(random) | 80.67 |
| +ThickOCR(complete) | 80.85 |
| +ThinOCR(random) | 81.19 |
| +ThinOCR(complete) | 85.03 |
| +LineEraseOCR(x-axis) | 85.84 |
| +LineEraseOCR(y-axis) | 85.31 |
| +ElongateOCR(x/horizontal) | 78.66 |
| +ElongateOCR(y/vertical) | 66.55 |

Table 3: Test accuracy(%) on the handwritten image augmentation methods using EMNIST dataset.

character, the neural network can learn better. Similar to the people handwriting, the same letter is written different by two people. Based on the pen pressure, their style, the curve they make in each stroke of the character, the stroke length, angle everything changes as notified by the signature verification systems. This proposed augmentation technique bring the similar strategy without altering the character entirely and can develop a better dataset.

ThickOCR, ThinOCR, LineErasingOCR and ElongateOCR methods are applied to only 50% of the dataset similar to Dropout. This results in performance increase. We alter the ThinOCR(random) probability to 0.2% to know whether the performance of the model changes. From the results, it is clear that accuracy of the model increased when compared to probability 0.5%. From the table 4, the accuracy of the model increases but only upto an extent.

| OCRThin(random) | p(0.2) | p(0.5) |
|---|---|---|
| MNIST | 99.06 | 99.00 |
| KMNIST | 92.20 | 92.04 |
| EMNIST | 81.17 | 81.19 |

Table 4: Test accuracy(%) on the handwritten image augmentation methods using OCRThin(random). The probability shows among the 50% of the data, how much each image is affected when augmentation.

## 6 Limitations

The limitations of the proposed data augmentation method is that this works only for handwritten characters. Although the meaning of the dataset is not changed with the proposed augmentation techniques, there exists languages which contains a single dot in the character. For example, in English the character 'i' has a dot representing the character. If the dataset contains a very small pixel area which represents the dot, there is a possibility of erasing the dot when ThinOCR or LineEraseOCR augmentation is done on the dot. This may result in changing the meaning of the character from 'i' to character 'l'. This limitation happens if the random value chosen by LineEraseOCR lies on the pixel. Although it affects the random image within the class, the effect can be negligible for huge datasets.

## 7 Conclusion

Data augmentation has been used to develop promising ways to increase the accuracy for classsification tasks. The traditional augmentation technique is very effective in image datasets, other techniques like color augmentation, GANs looks promising. We experimented with our own simple way of developing close to similar handwritten image datasets which GANs can create easily if used for same purpose. In future with more documented datasets, it is possible to develop a better baseline classification model for handwritten datasets before tuning the parameters or implementing complex models.

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
