# OpenReview forum: "Handwritten stroke augmentation on images"
_TMLR — Withdrawn by Authors_

### Review · Reviewer_mqCr · 2022-03-30

**Summary Of Contributions:**

This paper has tried four simple data augmentation methods on three handwritten text recognition datasets. The results are mostly not effective but there are a few cases where the proposed methods provide some gains.

**Requested Changes:**

* A major revision would be needed to improve the quality of the paper.
* It needs to clearly tell the problems to be solved in the paper and show evidences supporting the proposal.
* More extensive experiments will need to be done to show the effectiveness of the approach including comparisons with prior studies.
* It would be valuable to run experiments on non-single character handwritten datasets such as IAM.

**Strengths And Weaknesses:**

**Strengths**

I do not see noticeable strengths in the paper.

**Weaknesses**

1. The paper is really hard to read. Both contents and the quality of writing need to be improved. The scientific problem to be solved in the paper is not well introduced. It is hard to understand the details of the proposed methods. The figures do not tell meaningful things.

2. The results are not convincing. There is a small gain for Kuzushiji MNIST and there are some gains for EMNIST. However, it is not clear if the improvements are generally valuable. The model is too weak to assess the significance of the method. The methods are not compared with any previous augmentation methods for texts.

---

### Review · Reviewer_Ze81 · 2022-04-09

**Summary Of Contributions:**

The paper proposes image-based augmentation techniques for handwritten character images in the context of optical character recognition (OCR). It applies straightforward augmentations to images of single characters, involving increasing or decreasing the stroke thickness or removing a line horizontally or vertically. The paper evaluates the effectiveness of these approaches by training a small network on the MNIST, EMNIST (letters in English), and KMNIST (Japanese characters).

**Broader Impact Concerns:**

There are no ethical implications.

**Requested Changes:**

The paper requires a major revision to address the actual OCR task or extend the scope of the work. For the former, the IAM-OnDB (for the unstructured) or the Deepwriting datasets (an extended and character-wise labeled version of the IAM-OnDB for the structured settings) can be used. If the proposed augmentation techniques are the main contribution of the paper, then they should be compared against other techniques such as affine transformation.

The augmentation techniques can also be evaluated by using different architectures with varying capacities. It is rather inconclusive if the performance improvement is due to augmentations or the underlying model's capacity.

**Strengths And Weaknesses:**

Data augmentation is one of the established components in the deep learning pipeline and is frequently applied. It is known that it generally helps models learn more generalizable representations. However, I do not think that this paper is interesting for the community considering the scope of the proposed augmentation techniques and the scale of the experiments.

There is an inconsistency between the motivation and the scope of the paper. It is motivated by the OCR task, yet it ignores the in-the-wild OCR literature (i.e., the unstructured setting as defined in the paper). While datasets like MNIST, EMNIST, and KMNIST are valuable for validating theories or fundamental contributions, I do not think they are challenging enough for the OCR task.

Furthermore, the presented techniques are not motivated. Is there a particular reason why they are selected? Standard augmentation techniques such as affine transformation could be applied as well, which would cover the variations in handwritten text better.

I find the performance gain on the EMNIST dataset (from 77.99 to 85.84) with the proposed augmentations significant and rather unexpected. The proposed augmentations barely change the original sample. How can they be so effective on a larger dataset?

---

### Review · Reviewer_dhqU · 2022-04-11

**Summary Of Contributions:**

The paper proposes a set of (four) data augmentation techniques for handwritten characters (of potentially any language) in order to achieve better generalization in downstream tasks. The augmentation methods are claimed to be inspired by signature recognition literature (where different properties of strokes are important) and focus on tweaking “strokes” in the characters. It has also been claimed that the method ".. yields better performance of models for handwritten image datasets". Three datasets (MNIST, KMNIST, EMNIST) have been used for validating the methods on image level classification tasks.

**Broader Impact Concerns:**

Not applicable for this paper.

**Requested Changes:**

A LOT of changes would be needed for the paper to be considered acceptable.

- Entire rewrite of introduction and literature sections without unrelated content and focusing on the core topic of interest.
- Well-written algorithmic descriptions.
- Proper rationale behind the design of the algorithms are must.


**Strengths And Weaknesses:**

Overall comment: The paper, overall, solves only a small problem; that too without proper scientific (theoretical/empirical) evidance. There is virtually no explanation behind why the algorithms have been designed the way they are. Moreover, all the algorithms are mostly just simple image processing algorithms which existed for a long time.

The writing quality is also not up to the mark and description of the core algorithms are improper and sometimes incomplete. Following specific comments I would like to make.

- Proper algorithmic descriptions (with mathematical notations) must be provided for the core augmentation methods. At the current form (only textual descriptions), it is unclear what the exact steps are.
- The qualitative examples provided in the paper are quite unintelligible. I could barely notice the difference between the different columns of Fig. 2, 3 & 4.
- ThinOCR and ThickOCR are sort-of similar to standard morphological operations. No comparison OR discussion have been done.
- The “Introduction” and “Literature” section argues why these methods are inspired by Signature Verification tasks and focused on “strokes” (and their properties like thickness etc). This do not hold in the actual algorithm – e.g. “LineEraseOCR” is essentially a matrix/grid level operation and nothing to do with strokes.
- The introduction and literature sections are full of unnecessary discussions which are not relevant to the method in question. Literatures for signature verification have been provided whereas there is hardly any significant connection between them and the proposed augmentation methods.
- Importantly, results for MNIST are not improving with the proposed method. No explanation has been provided. Also, why is the dropout reducing test accuracy?

Some clarifications needed:

- Abstract calls the method "data-driven" -- what does that exactly mean ? Isn’t it just image processing operations?
- What do the authors mean by “structured/unstructured” and “well documented” (para 4 of introduction) datasets?

---

### Review · Reviewer_NYDc · 2022-04-12

**Summary Of Contributions:**

In this paper, the authors propose 4 methods to perform data augmentation for training character recognition models. These methods consist of local modifications of the pixels of  handwritten character stokes. The methods are used to train a convolutional neural network, which is then evaluated on 3 isolated handwritten character datasets (MNIST, EMNIST and KMNIST).

**Requested Changes:**

- give state-of-the art performance on the three databases
- compare the proposed methods to other data augmentation methods
- consider a more complex problem such as handwritten text recognition for which many databases and recognition systems are freely available

**Strengths And Weaknesses:**

Limitations:
- The isolated character recognition task is a "toy" problem that has no real application. The handwriting recognition community is now focusing on the real tasks such as recognising lines of handwritten text or even complete pages.
- The proposed methods do not seem to extend to cursive handwriting recognition (lines or pages)
- The proposed methods require a binarised image where the black pixels are considered as belonging to the character. However, in real cases, e.g. for historical documents, binarisation is not straightforward and it is better to perform the recognition directly on the greyscale image. State-of-the-art handwriting recognition systems do not include a binarisation step.
- The methods proposed by the authors are not compared to other methods proposed for OCR[1] or NLP[2].
- The performance of the base model is not compared to the performance of state-of-the-art models on the bases used.
In conclusion, the experiments conducted by the authors are not sufficient to demonstrate the interest of their methods, in particular no comparison is made with the state of the art. The scope of the proposed methods is also limited to the recognition of isolated characters which is a simple problem.

---

### Note · Authors · 2022-04-15

**Comment:**

We thank all the reviewers for their inputs. We would like to withdraw our paper as our paper needs major revision and the concept is not strong enough. Valuable inputs received from the reviewers will be taken for our papers in the future.


**Withdrawal Confirmation:**

I have read and agree with the venue's withdrawal policy on behalf of myself and my co-authors.